# Synthesis of Nucleobase-Modified RNA Oligonucleotides by Post-Synthetic Approach

**DOI:** 10.3390/molecules25153344

**Published:** 2020-07-23

**Authors:** Karolina Bartosik, Katarzyna Debiec, Anna Czarnecka, Elzbieta Sochacka, Grazyna Leszczynska

**Affiliations:** Institute of Organic Chemistry, Faculty of Chemistry, Lodz University of Technology, Zeromskiego 116, 90-924 Lodz, Poland; karolina.bartosik@p.lodz.pl (K.B.); katarzyna.debiec@dokt.p.lodz.pl (K.D.); czarneckann2@gmail.com (A.C.); elzbieta.sochacka@p.lodz.pl (E.S.)

**Keywords:** phosphoramidite chemistry, RNA solid-phase synthesis, post-synthetic RNA modification, convertible nucleoside

## Abstract

The chemical synthesis of modified oligoribonucleotides represents a powerful approach to study the structure, stability, and biological activity of RNAs. Selected RNA modifications have been proven to enhance the drug-like properties of RNA oligomers providing the oligonucleotide-based therapeutic agents in the antisense and siRNA technologies. The important sites of RNA modification/functionalization are the nucleobase residues. Standard phosphoramidite RNA chemistry allows the site-specific incorporation of a large number of functional groups to the nucleobase structure if the building blocks are synthetically obtainable and stable under the conditions of oligonucleotide chemistry and work-up. Otherwise, the chemically modified RNAs are produced by post-synthetic oligoribonucleotide functionalization. This review highlights the post-synthetic RNA modification approach as a convenient and valuable method to introduce a wide variety of nucleobase modifications, including recently discovered native hypermodified functional groups, fluorescent dyes, photoreactive groups, disulfide crosslinks, and nitroxide spin labels.

## 1. Introduction

Recently, a significant interest has been observed in the use of modified oligoribonucleotides in the fields of molecular biology, biochemistry, and medicine [1,2,3]. The representative sites of RNA modifications are the 5′- and 3′-ends of oligoribonucleotides, and the 2′-positions of the ribose, phosphodiester residues, and nucleobase moieties. Notably, among 150 modified nucleosides identified in cellular RNA sequences, more than 95% of functional groups are installed in the nucleobase heterocycles (Figure 1) [4,5]. Improved mass spectrometry approaches and highly sensitive chemical screens in cellular RNA isolation are still extending this list with new base-modifying units, e.g., cyclic *N*^6^-threonylcarbamoyladenosines (ct^6^A, ms^2^ct^6^A) [6,7], 2-methylthiomethylenethio-*N*^6^-isopentenyl adenosine (msms^2^i^6^A) [8], uniquely lipophilic 5-substituted 2-geranylthiouridines (mnm^5^ges^2^U, cmnm^5^ges^2^U) [9] and 2-methylthiocytidine (ms^2^C) [10]. RNA modifications affect the structure, stability, and biological activity of RNA biomolecules, particularly the translation process [11]. Experiments of the site-specific incorporation of nucleic base modifications have proved to be a particularly useful method for precisely assessing the role of various functional groups in the structure, function, and biosynthesis of RNA molecules [12,13,14,15,16,17]. 

Valuable data on RNA structure and cellular activity can be also extracted from biochemical, biophysical, and structural studies involving unnatural base-modified RNA constructs. The insertion of artificial purine/pyrimidine in the place of naturally existing nucleosides may change the local hydrogen bonding system, stacking interactions, or puckering of ribose residues, giving answers as to which modifications or interactions are essential for the functional RNA structure [18,19,20]. Important details on RNA structure, conformational dynamics, and RNA’s homo- and heterotropic interactions can be provided by oligoribonucleotides site-specifically labeled with fluorescent dyes, nitroxide radicals, disulfide crosslinks, or photolabile groups [21,22,23,24,25,26,27,28]. 

For years, chemically modified oligonucleotides have been extensively studied for the development of new antisense and siRNA therapeutics [1,29,30,31]. Several nucleobase modifications were found to improve the drug-like properties of oligonucleotides such as cellular uptake, stability in the cell, target specificity, and binding affinity [31,32,33], and reduce undesirable protein binding and immune stimulation [34,35]. Gratifyingly, the incorporation of the first-generation internucleotide linkage modifications and second-generation sugar modifications resulted in the current approval of two RNAi-based therapeutics, Onpattro (patisiran) and Givlaari (givosiran) [31].

Growing interest in the modified RNA oligomers creates the need to develop chemistry that permits the site-specific introduction of functionalizable groups into RNA. The most general way to address the issue of site selectivity is through the use of solid-phase synthesis phosphoramidite chemistry, which involves two strategies: (1) classical modified monomer approach via the preparation of a modified phosphoramidite building block and its subsequent incorporation into the RNA chain; and (2) a post-synthetic RNA modification approach based on selective chemical reaction(s) of precursor unit(s) in the full-length oligonucleotide prepared by the classical method. 

This review for the first time summarizes the methods of site-specific incorporation of nucleobase-modified units into RNA oligomers via the post-synthetic strategy. We focused on the nucleobase functionalization due to the high abundance of nucleobase modifications in the nature and the high convenience of the post-synthetic strategy to construct the nucleobase-labeled chemical or biophysical probes. The scope of post-synthetically reactive precursor oligonucleotides has been limited to RNA oligomers prepared by the phosphoramidite chemistry. To facilitate the selection of post-synthetic methods best suited to the assumed chemical–biological objectives, a large number of experimental details have been included. 

## 2. Solid-Phase Synthesis of Modified RNA Oligomers via Phosphoramidite Chemistry

The most current method for the incorporation of a modified nucleoside into a site-specific position of oligoribonucleotides is the phosphoramidite chemistry, which was introduced by Caruthers in 1981 [36,37]. The preparation of RNA oligomers by this strategy involves a four-step reaction cycle including 5′-deblocking, coupling, capping, and oxidation steps (Scheme 1). The synthesis takes the 3′→5′ direction and starts from the 5′-deprotection of fully protected ribonucleoside attached to a solid support (in standard, controlled-pore glass, or polystyrene resins) via the 3′-hydroxyl group. In the second step, activation and coupling, the nucleoside deprived of the 5′-protecting group reacts with a suitably protected phosphoramidite building block activated by the 5-substituted 1*H*-tetrazole or imidazole derivatives, yielding an unstable phosphite triester linkage. Since a small number of unreacted 5′-hydroxyl sites remain active after coupling, in the third step, capping, the acylating reagent is introduced to prevent them from reacting with the next monomeric unit and elongating the missense strands. In the oxidation step, the support-linked dinucleoside phosphite triester P(III) is converted to the more stable phosphate triester P(V) with an oxidation agent, typically iodine or *tert*-butylhydroperoxide solutions.

The removal of the 5′-protecting group initiates the next chain extension cycle. Repeating the synthetic cycle provides a full-length oligonucleotide (prepared in the trityl-on or trityl-off mode), which can be released from the solid support and deprotected. 

For successful solid-phase RNA synthesis, the phosporamidite monomeric units must be protected with a combination of orthogonal R_1_ transient and R_2_, R_3_, R_4_ permanent protecting groups. Several synthetic protocols, systems of protecting groups, and deprotection strategies have been demonstrated so far [38]. In practice, the building blocks are protected by employing one of the five most common strategies: (1) 5’-*O*-DMTr-2’-*O*-TBDMS [39,40,41]; (2) 5’-*O*-DMTr-2’-*O*-TOM [42]; (3) 5’-*O*-DMTr-2’-*O*-Fpmp [43]; (4) 5’-*O*-DMTr-2’-*O*-TC [44], and (5) 5’-*O*-DOD(BzH)-2’-*O*-ACE [45,46] (Figure 2). 

Most of the synthetic approaches involve 3′-*O*-*β*-cyanoethyldiisopropylphosphoramidites containing the transient, acid-labile 5′-*O*-dimethoxytrityl group (DMTr) and permanent 2′-protection such as silyl blockage (TBDMS or TOM, Figure 2a,b), the acetal masking group (Fpmp, Figure 2c) or carbothioate (TC, Figure 2d). The deprotection of trityl-off RNA oligomers prepared via TBDMS/TOM and Fpmp chemistry starts from the removal of base-labile protecting groups from the phosphotriester backbone and exoamino functions of nucleobases (usually acyl or aminomethylene blockage (Figure 2g). Typically, ammonolytic conditions are used such as aqueous (aq.) ammonia or aq. ammonia–methylamine solution (AMA). This alkaline step of deprotection runs simultaneously with the support cleavage. Optionally, the removal of *β*-cyanoethyl phosphate groups can be performed separately before the alkaline deprotection step using a weak base in an organic solvent (e.g., TEA/acetonitrile, 1:1 *v/v* or 10% diethylamine in acetonitrile). Separation of the released acrylonitrile by-product prevents the formation of 2-cyanoethyl-containing oligomer adducts observed when the strong basic conditions of RNA deprotection are used. Removal of the 2′-protecting groups is performed as a final step of oligomer deprotection in the presence of fluoride agent (TBAF or TEA⋅3HF) for 2′-silyl groups or under acidic conditions (AcOH) when 2′-Fpmp acetal blockage is used. The deprotection of the TC-blocked support-bound oligonucleotide is carried out on a synthesizer column in one step by filling the column with neat ethylenediamine. After TC removal, the “free” oligomer is washed from the column using a small volume of water. 

An alternative approach for the orthogonal protection of the 5′- and 2′-hydroxyl groups of monomeric units (prepared as methyl diisopropylphosphoramidites) is based on fluoro-labile 5′-*O*-silyl protecting groups (DOD or BzH) and acid-labile 2′-*O*-bis(2-acetoxyethoxy)methyl ortoester protection (ACE, Figure 2e) [45,46]. The exocyclic amino functions of nucleobases are masked with base-labile acyl protecting groups. After 5′-*O*-DOD/BzH removal (HF/TEA), the oligomer is subjected to a three-step deprotection protocol involving the release of phosphate groups by disodium-2-carbamoyl-2-cyanoethylene-1,1-dithiolate (S_2_Na_2_), the removal of base-labile protecting groups, including acetyl groups from 2′-*O*-ACE blockage using methylamine solution with simultaneous resin cleavage, and finally, the removal of acid-labile 2′-*O*-bis(2-hydroxyethoxy)methyl orthoesters with AcOH/TEMED. 

In addition to the classical phosphoramidite approach proceeding in a 3′→5′ direction, interest has also been focused on the RNA synthesis in the reverse direction (5′→3′) [47,48]. This approach utilizes 5′-*O*-phosphoramidites (Figure 2f) with a suitable *N*-protecting group (Bz for adenine, cytosine, Ac for cytosine, and iBu for guanine) and a 3′-*O*-DMTr-2′-*O*-TBDMS blockage of sugar residue. Activated 5′-*O*-phosphoramidite is coupled with a nucleoside bearing a 5′ succinate of ribonucleoside attached to a solid support leading (after oxidation) to the dimer formation with phophodiester linkage and a 3′-*O*-DMTr-protecting group. The removal of DMTr opens the next synthetic cycle. Oligonucleotide deprotection is performed using the same standard TBDMS chemistry currently utilized in the conventional 3′→5′ synthesis. Although the reverse RNA synthesis is used to a lesser extent than the conventional strategy, it offers a facile route to the assembly of 3′-conjugated RNA constructs of long and medium lengths. Importantly, it was demonstrated that the use of 3′-*O*-DMTr-5′-phosphoramidites enhances the coupling efficiency, which surpassed 99% per step leading to high-purity RNA products [48]. These distinct advantages of 5′→3′ direction synthesis made the reverse 3′-*O*-DMTr-5′-phosphoramidites commercially available.

After deprotection, synthetic oligomers (obtained in both conventional and reverse strategies) are purified using polyacrylamide gel electrophoresis (PAGE) or high-performance liquid chromatography (RP HPLC, IE HPLC) and identified by electrospray ionization mass spectrometry (ESI), matrix-assisted laser desorption/ionization-time of flight mass spectrometry (MALDI-ToF), and enzymatic digestion analysis.

The solid-phase phosphoramidite method is limited to synthesizing RNA oligomers with lengths of 50–100 nucleotides (longer RNAs molecules can be synthesized by the enzymatic ligation of chemically modified RNA fragment(s) using T4 DNA or T4 RNA ligases [49,50,51] or DNA-zymes [52]). In response to the clinical and commercial success of the therapeutic oligonucleotides, the classical phosphoramidite protocols were translated for use in the large-scale synthesis [53]. Using automated computer-controlled synthesizers, the incorporation of several modifications (phosphorothioate internucleotide linkage, 2′-*O*-methoxyethyl, 2′-*O*-Me, 2′-F, locked-, morpholino- and Spiegelmer building blocks) was scaled up from gram- to kilogram-scale production under Current Good Manufacturing Practice (cGMP). 

## 3. Post-Synthetic Strategy for Nucleobase RNA Modifications 

First protocols for the post-synthetic functionalization of nucleobases were elaborated for DNA oligomers about three decades ago [54,55,56,57,58,59]. The general concept of post-synthetic RNA modification approach relies on the preparation of an easily convertible/reactive precursor oligonucleotide by the chemical method, e.g., phosphoramidite chemistry, and its subsequent derivatization through chemical reaction(s) involving the reactive center(s) of precursor nucleoside(s). The representative sites of post-synthetic modifications are 5′- and 3′-ends of RNA oligomers, the 2′-position of sugar, phosphate linkage, and nucleobase moieties (Figure 3). The derivatization of nucleobases attends the C-4, C-5, and N-3 positions of uracil, the C-5 position of cytosine, the C-2 position of adenine, and the C-8 and N-7 sites of guanosine (Figure 3). Some of the functional groups have also been attached to the exocyclic amino functions of C, A, and G as well as the thiocarbonyl group of thiouridines (Figure 3).

Post-synthetic conversions of RNA oligomers can be performed in the solid or liquid phase (Scheme 2). The “in solid-phase” approach involves a fully protected, support-attached oligoribonucleotide as a substrate for the post-synthetic reaction (Scheme 2A). After conversion, the oligomer is released from the resin and deprotected. Notably, the use of basic conditions for the transformation process can promote the simultaneous removal of exoamino and phosphate-protecting groups as well as the support cleavage. In these cases, the post-synthetic reaction is carried out in a one-step conversion–deprotection process. Occasionally, the separate deprotection of the phosphate groups is performed prior to post-synthetic conversion to avoid any side reactions of the oligomer with acrylonitrile released during removal of the *β*-cyanoethyl groups.

Alternatively, the post-synthetic RNA modification can be carried out in the liquid phase (Scheme 2B). This “in-solution” approach involves a fully deprotected and released precursor oligoribonucleotide. After purification, the “free” oligomer is converted to the target product and repurified to remove the excess of reagents. It is imperative that the presence of free 2′-hydroxyl groups excludes the use of highly alkaline conversion conditions that could promote the strand cleavage or phosphate migration.

Correct incorporation of the final modified unit should be verified by careful analysis of the isolated product by MALDI-ToF, ESI-MS, and/or enzymatic digestion.

The post-synthetic strategy of RNA modification has several important advantages in comparison to the classical modified monomer approach. Firstly, one single modified precursor oligoribonucleotide can provide several sequentially homologous, variously modified RNA fragments [60,61,62]. Such a strategy significantly reduces the cost of synthesis compared to preparing each modified monomeric unit separately. Secondly, hypermodified monomeric units containing highly reactive groups e.g., –COOH, -SO_3_H, -CHO can be replaced by structurally convenient, easy convertible precursor phosphoramidites deprived of problematic functional groups [62]. Finally, the post-synthetic strategy permits introducing nucleosides/labels that are sensitive to the conditions of solid-phase synthesis and/or oligomer deprotection [63,64,65]. In some cases, the use of post-synthetic protocol proved to be the method of choice [63,64]. 

The key to an effective post-synthetic RNA modification strategy is the reactivity of the precursor nucleoside, which should ensure almost quantitative conversion without any perturbation of RNA structure. In addition, the precursor nucleoside should offer an easy approach to afford the phosphoramidite building block and its effective incorporation into the RNA chain. Therefore, the choice of precursor compounds is limited to nucleosides that are fully compatible with the protocols of RNA synthesis and deprotection. 

The limited stability and hydrophilic character of oligoribonucleotides significantly reduce the number of organic reactions that can be used in post-synthetic transformations of precursor RNA oligomers. In practice, the conditions of post-synthetic reactions are restricted to polar solvents, moderate temperatures (less than 60 °C), and reaction times shorter than 24 h. Despite the above-mentioned restrictions, the conversions attending the heterocyclic bases represent the highest chemical diversity demonstrated by several types of reactions, including nucleophilic aromatic substitution, allylic substitution, carbon–carbon bond-forming reaction via Sonogashira and Stille couplings, cycloaddition reactions, the derivatization of aliphatic amino groups by the formation of amide bonds, and finally, the functionalization of thiouridines via desulfuration, the formation of thioether bonds, and disulfide bridges. All of the above-mentioned post-synthetic conversions are discussed in the following chapters.

### 3.1. Nucleophilic Aromatic Substitution

Nucleophilic aromatic substitution (S_N_Ar) is the most common post-synthetic reaction utilized in the synthesis of nucleobase-modified oligoribonucleotides. It involves a convertible RNA oligomer equipped with a nucleoside containing a good leaving group(s) directly attached to the heterobase and easily displaced by nucleophile(s). The post-synthetic S_N_Ar reaction enables synthesizing single- or multiple-modified target oligonucleotides, including double-derivatized products yielded by the attachment of bifunctional nucleophile and designed for further derivatization with a chemical probe. The literature provides several examples of convertible ribonucleosides subjected to post-synthetic S_N_Ar transformations (Figure 4). Most of them contain aryl ether leaving groups (**1**–**3**, **6**) directly attached to the C4 position of pyrimidine and the C6 position of purine heterobases. Others contain a triazolyl group directly linked to the C4 position of the pyrimidine ring (**5**) or sulfonate residue installed in the C6 position of purine nucleoside (**7**). The representative leaving groups are also halides attached to the C5 atom of uridine (**4**) and C2 or C6 atoms of purine nucleosides (**8**, **9**).

The first convertible oligoribonucleotides were reported by Verdine and co-workers in 1995 [66,67]. The post-synthetic strategy attended the exocyclic amino groups of A, C, and G nucleosides and was an extension of earlier Verdine’s lab studies on the convertible nucleoside approach in the derivatization of DNA oligomers [54,55,57,58,68]. Developed chemistry permits the site-specific incorporation of *N*-functionalizable nucleobases of A, C, and G into RNA and its further derivatization to e.g., crosslinked nucleic acids (see Scheme 27A). The authors demonstrated the use of 4-*O*-(4-chlorophenyl)uridine (**1**, see Figure 4), 6-*O*-(4-chlorophenyl)inosine (**6**), and 2-fluoro-6-*O*-(4-nitrophenylethyl)inosine (**9**) prepared as 5′-*O*-DMTr-2′-*O*-TBDMS-protected phosphoramidites to the synthesis of 11-nt precursor oligonucleotides **10**–**12** by standard 1 µmol phosphoramidite chemistry (Scheme 3). For use in the automated RNA synthesis, the convertible Verdine’s phosphoramidites were applied as 0.1 M solutions in acetonitrile, while the coupling time was extended to 12 min. Syntheses proceeded with average stepwise yields of ca. 97%. The following conversion–deprotection step of resin-bound oligonucleotides with primary alkylamines resulted in the substitution of chlorophenyl groups in aryl ethers (oligomers **10, 11**) as well as 2-fluoride leaving groups in *O*^6^-protected inosine (oligomer **12**) with amines providing *N*^4^-alkylcytidine-(**14a**–**h**), *N*^6^-alkyladenosine-(**15a**–**h**) and *N*^2^-alkylguanosine-modified RNAs (**16a**–**h**). To eliminate the base-promoted RNA degradation, the post-synthetic transformations were performed under anhydrous conditions using 7 M methanolic ammonia, 8 M solution of methylamine in ethanol, or 2 M methanolic solutions of other amines such as ethylenediamine, 1,4-diaminobutane, cystamine, ethanolamine, benzylamine, and 2-(methylthio)ethylamine. Reactions were performed at 42 °C for 12–18 h. After displacement, the mixture of oligomer and remaining amine reagent was separated from the support and then evaporated or eluted through a Dowex cation exchange column (NH_4_^+^ form). Subsequent desilylation involved TEA·3HF or 1 M TBAF/THF treatment. Target oligonucleotides were purified by denaturing PAGE, affording oligomers **14**–**16** in the relative conversion yields above 65%. Measurements of thermal stability of modified duplex RNA oligomers have revealed that in most cases, the presence of *N*-alkyl modification slightly reduces duplex stability. The exceptions were oligonucleotides carrying a tether with a “free” amine group (oligomers **14c**–**e**, **15c**–**e** and **16c**–**e**) for which increased duplex stability was observed. 

Currently, the convertible phosphoramidites introduced by Verdine and co-workers are commercially available as 5′-*O*-DMTr-2′-*O*-TBDMS-protected building blocks.

Based on the Verdine’s strategy, Babaylova and co-workers introduced an ethylenediamine linker to the C5 position of uridine employing 5-bromouridine **4** (see Figure 4) as a convertible nucleoside [69]. The CPG-bound 5′-*O*-DMTr-2′(3′)-*O*-acetyl-C5-bromouridine was employed for the synthesis of a 10-nt RNA oligomer **13** (Scheme 3) at the 0.4 µmol scale followed by treatment with 2 M solution of ethylenediamine in methanol (42 °C, 18 h). After the desilylation process, amino-oligomer **17c** was isolated by denaturating PAGE electrophoresis. The attachment of a bifunctional ethylenediamine nucleophile in this fashion provided a convenient substrate for further functionalization with nitroxide radicals.

The next exploration of Verdine’s ribonucleosides **1**, **6**, and **9** (see Figure 4) was demonstrated by the Höbartner group to install the nitroxide spin labels on exocyclic amino groups of RNA nucleobases of C, G, and A (Scheme 4 and Scheme 5) and evaluate the potential of TEMPO- and proxyl-labeled nucleobases for probing RNA secondary structures by PELDOR experiments [52,70], and construct, among others, riboswitches of the correct regulatory function [52]. The commercially available 5′-*O*-DMTr-2′-*O*-TBDMS-protected Verdine’s phosphoramidites were incorporated into RNA by solid-phase synthesis using a polystyrene support in combination with 5′-*O*-DMTr-2′-*O*-TOM-protected canonical monomeric units (products were prepared as single or double-modified 12-, 13- or 20-nt oligomers) [70].

The support-bound, fully protected precursors **10** and **12** were incubated with 2 M solution of 4-amino-2,2,6,6-tetramethylpiperidin-1-oxyl (TEMPO-NH_2_, **18a**) in methanol at 42 °C for 24 h providing fully converted *N*^4^-TEMPO-cytidine- and *N*^2^-TEMPO-guanosine-containing RNAs, respectively. However, the same transformation of *O*^6^-(4-chlorophenyl)inosine-oligomer **11** to *N*^6^-TEMPO–adenosine–RNA turned out incomplete even after a prolonged incubation time and temperature. All nucleobase substitutions were performed in one step with support cleavage and the deprotection of base-labile protecting groups. After desilylation, the TEMPO-substituted RNA oligonucleotides **19**–**21** were purified by anion-exchange HPLC or denaturing PAGE. The influence of TEMPO subsituents on RNA conformation and the secondary structure of the hairpin and duplex structure was investigated, demonstrating the potential of labeled oligomers as reporter probes in pulsed EPR spectroscopy. Although the resulting spin-labeled RNAs were obtained in good yields and high qualities, further work was undertaken to improve the preparation of the precursor oligomers. To this purpose, Verdine’s 4-*O*-(p-chlorophenyl)uridine building blocks protected with 2′-*O*-TOM or 2′-*O*-Me group were used in the place of commercially available 2′-*O*-TBDMS phosphoramidite (Scheme 5) [52]. This change significantly improved the coupling yields and shortened coupling times. In addition, 2′-OMe modification was expected to increase the stability of nitroxide-labeled oligomers. For post-synthetic transformations, fully protected, polystyrene-linked oligomers **22**, **23** were incubated with a 2 M methanolic solution of TEMPO-NH_2_ (**18a**) or 3-amino-2,2,5,5-tetramethylpyrrolidin-1-oxyl (proxyl, **18b**) at 42–55 °C for 24–48 h. Then, aq. MeNH_2_ or aq. NH_3_ were added to complete the cleavage of base-labile protecting groups (this step was skipped during the preparation of short TEMPO-labeled oligomers, which were deprotected simultaneously with displacement). Fully deprotected TEMPO- (**24a**, **25a**) and proxyl-labeled oligomers (**24b**, **25b**) ranging from 9 to 27 nt were purified by PAGE or anion-exchange HPLC. For proxyl-labeled RNAs **24b**, **25b**, the formation of an *N*^4^-methylcytidine-containing side product was observed, resulting from the substitution of the 4-*O*-chlorophenyl group in an unreacted precursor oligomer with methylamine. The nitroxide-labeled RNAs carrying TEMPO or proxyl were utilized to obtain functional 53-nt and 118-nt riboswitch RNAs by deoxyribozyme-catalyzed ligation as an alternative to protein-based RNA ligation [52]. 

Commercially available 5′-*O*-DMTr-2′-*O*-TBDMS Verdine’s phosphoramidites were applied for the incorporation of photolabile 2-(2-nitrophenyl)propyl (NPP) and 2-(2-nitrophenyl)ethyl (NPE) “caging groups” (Scheme 6) [71]. Solid-phase syntheses of 15-nt precursor oligomers **10**–**12** were performed according to standard coupling protocol. For displacement reaction, fully protected and support-bound oligomers **10**–**12** were treated with 2.5 M NPP-NH_2_ (**26a**) in methanol (1–2 days, 25–42 °C) or NPE-NH_2_ (**26b**) in acetonitrile (2–5 days, 42–60 °C). The use of sterically hindered amine **26b** required an extra cleavage step with methylamine or aq. NH_3_ for complete base deprotection. In the case of inosine-containing oligomer **11** the conversion with NPE–NH_2_ turned out not to work. Finally, oligomers containing both NPP and NPE groups were desilylated using TEA⋅3HF/TEA/NMP mixture. An attempt to prepare 4-*O*-caged uridine- and 6-*O*-caged adenosine-containing oligomers by incubation of **10**, **11** with NPP–OH nucleophilic reagent (alcohol analog of NPP–NH_2_) in MeCN for 7 days at 42 °C (optionally in the presence of tertiary amine) proved to be ineffective. The presented methodology enabled simplifying the synthesis of caged RNAs (compared to the standard monomer approach), since the commercially available Verdine’s phosphoramidite was used. The expected results of duplex destabilization caused by attached caging groups were achieved.

In 2003, Kierzek’s group employed 6-methylthiopurine riboside **7** (see Figure 4) and 2-methylthio-6-chloropurine riboside **8** as convertible nucleosides for post-synthetic nucleophilic aromatic substitution with primary alkylamines, leading to *N*^6^-alkyladenosine- and 2-methylthio-*N*^6^-alkyladenosine-modified oligoribonucleotides (7-, 8- and 17-mers), respectively (Scheme 7 and Scheme 8) [60]. Among them, four naturally occurring tRNA modifications i^6^A, m^6^A, ms^2^i^6^A, and ms^2^m^6^A were inserted. Both convertible oligomers **30** and **34** were synthesized automatically on 1 µmol scale by the standard protocol of phosphoramidite chemistry on solid support (CPG, Fractosil 500 or resins loaded with 5′-*O*-DMTr-2′-*O*-TBDMS-protected convertible nucleosides **7**, **8**). Before conversion, the 6-methylthio group (6-SCH_3_) of support-linked 6-methylthiopurine-modified oligoribonucleotide **30** (Scheme 7) was selectively and quantitatively activated by oxidation with a 20 mM solution of monoperoxyphtalic acid (magnesium salt form, 2.5 h, rt) yielding a mixture of oligomers **31** and **32** decorated with 6-methylsulfoxide (6-S(O)CH_3_) and 6-methylsulfonyl (6-S(O)_2_CH_3_) groups, respectively. The oxidized oligomers **31**, **32** (still attached to the support) were reacted with various primary alkylamines. To get oligomers modified with *N*^6^-methyladenosine or *N*^6^-isopentenyladenosine, 2 M MeNH_2_ in THF (12 h, rt) and isopentenylamine hydrochloride with TEA in pyridine (12 h, 55 °C) were applied, respectively. Other amines, such as isoamylamine, neopentylamine, 1-methylpropylamine, 1-methylbutylamine, and propargylamine were used as a solution in acetonitrile (1:9 *v*/*v*) and left for 12 h at rt. After conversion, the samples were evaporated, and the aq. NH_3_/EtOH (3:1 *v*/*v*, 8 h, 55 °C) was added for the complete removal of the base-labile protecting groups. After desilylation, the fully deprotected oligomers were purified on silica gel plates. The overall yields of *N*^6^-alkyladenosine-containing RNAs **33a**–**h** were in the range of 40–77% (13–34% for 17-mers). It was indicated that the synthesis yields were not dependent on the position of convertible nucleoside in the RNA sequence but on the steric hindrance and the nucleophilic character of the amino group in alkylamine conversion reagents. 

To prepare the 2-methylthio-*N*^6^-alkyladenosine-modified RNAs **35a**–**c** (Scheme 8), several 6-substituted purine ribosides such as dinitrothiophenyl-, pentafluorophenyl-, and chloro-substituted purines were tested as model convertible nucleosides [60]. Among them, 2-methylthio-*N*^6^-chloropurine riboside **8** (see Figure 4) exhibited the highest reactivity with primary amines. Therefore, the 5′-*O*-DMTr-2′-*O*-TBDMS-protected convertible phosphoramidite of **8** was introduced into RNA sequences (7-, 8- and 17-mers) on the 1 µmol scale of the oligonucleotide synthesis. The trityl-off support-linked precursor **34** (Scheme 8) was treated with alkylamines: 2 M MeNH_2_ in THF (2–5 h, 55 °C) or isopentenylamine hydrochloride/TEA in pyridine (2–5 h, 55 °C). For complete removal of base-labile protecting groups, the samples were additionally treated with aq. NH_3_/EtOH (3:1 *v*/*v*, 8 h, 55 °C). Displacement of the 6-chloro group with –NH_2_ involved the aq. ammonia and prolonged reaction time (16 h, 55 °C). After desilylation, the fully deprotected oligomers were purified on silica gel plates. The overall yields of 2-methylthio-*N*^6^-alkyladenosine-modified oligoribonucleotides **35a**–**c** range from 20% to 65% (16–24% for 17-mers).

The post-synthetic introduction of *N*^6^-alkylated adenosines and 2-methylthioadenosines elaborated by Kierzek’s group seems to be the method of choice for obtaining several homologous RNA oligomers, especially that the precursor synthons were stable during the synthesis of 3′-*O*-phosphoramidite/oligonucleotide and easily transformed to target *N*^6^-substituted adenosine-RNAs.

The post-synthetic S_N_Ar reaction has shown to be attractive for the incorporation of 4-thiouridine (s^4^U) into RNA oligomers. The preparation of s^4^U-RNAs by the classical approach requires a special protection for 4-thiocarbonyl function in s^4^U-phosphoramidite to avoid the desulfuration by-product produced during the oxidation step [72,73]. This inconvenience has been overcome by employing the convertible nucleoside approach. Several examples of precursor ribonucleosides were reported for s^4^U-RNA preparation, including 4-(1,2,4-triazol-1-yl)-2-pyrimidon-1-yl-*β*-D-ribofuranoside (**5**, see Figure 4) [74,75,76], 4-*O*-(2-nitrophenyl)uridine (**2**, Figure 4) [76], and 4-*O*-(4-nitrophenyl)uridine (**3**, Figure 4) [77]. 

The use of 4-(1,2,4-triazol-1-yl)-2-pyrimidon-1-yl-*β*-D-ribofuranoside convertible nucleoside **5** was demonstrated by Shah and Kumar [74,75] (Scheme 9A). The 5′-*O*-DMTr-2′-*O*-TBDMS-protected phosphoramidite of **5** was incorporated into 5- or 7-nt oligonucleotides using standard phosphoramidite chemistry with more than 98% coupling efficiency. The trityl-off, CPG-linked precursor **36** was converted to s^4^U-RNA by treatment with 10% thioacetic acid (CH_3_C(O)SH) in acetonitrile (12 h, rt). Then, 10% DBU in methanol (16 h, rt) was used to release the oligomer from the support and to remove the base-labile protecting groups. The 2′-*O*-TBDMS groups were cleaved with TBAF to give the final s^4^U-RNA **38**. 

Thioacetic acid reagent was also employed by Komatsu and co-workers to obtain s^4^U-derivative **38** by the conversion of 4-*O*-(4-nitrophenyl)uridine-containing oligomer **37** prepared using 5′-*O*-DMTr-2′-*O*-TBDMS chemistry (Scheme 9A) [77]. The conversion and base–deprotection steps were performed according to Shah’s procedure [74] in 94% yield. After desilylation and purification (RP and IE HPLC), the overall yield of s^4^U-RNA oligomer **38** was determined as 10%. The 4-nitrophenyl ether leaving group was also effectively displaced with methoxy substituent by the incubation of precursor oligomer **37** with 10% DBU/MeOH at 30 °C for 16 h (Scheme 9B). Strong alkaline conditions permitted the simultaneous cleavage of base-labile protecting groups and resin. Further desilylation (TEA·3HF/DMSO, 1:1 *v*/*v*) and HPLC-purification afforded 4-*O*-methyluridine-RNA **39** in an overall 10% yield.

In 2004, Avino and co-workers prepared s^4^U-RNAs by the post-synthetic conversion of oligomers containing 4-triazolyl-pyrimidine riboside **40** and 4-*O*-(2-nitrophenyl)uridine **41** with sodium hydrogen sulfide NaSH (Scheme 10A) [76]. To avoid oligomer degradation under base conversion conditions, the 5′-*O*-DMTr-2′-*O*-Fpmp-protection system was applied for monomeric units. The 6- and 10-nt RNA oligomers were synthesized on a 1 µmol scale using CPG support. The trityl-on support-linked precursors **40** and **41** were treated with a 0.2 M NaSH solution in DMF (3 h, rt) to achieve complete displacement of the triazolyl group. Then, concentrated ammonia was added (4 h, 50 °C) to release the oligomers from the support and remove the base-labile protecting groups. The obtained trityl-on 2′-protected-RNAs were purified by RP HPLC and then treated with 10 mM aq. HCl (pH 2.5–3, 12 h, rt), yielding ca. 10 OD of the final s^4^U-oligomer **42**. It was found that the triazolyl derivatives (phosphoramidite building block and precursor oligomer **40**) have a limited shelf life (<3 months), while the 4-*O*-(2-nitrophenyl)uridine derivatives appeared significantly more stable (shelf life > 3 years). 

The convertible approach based on the 4-triazolyl pyrimidine nucleoside **5**, which was initially introduced by Shah and co-workers [74] for s^4^U-RNA preparation (Scheme 9), was adopted by Guennewig [78] and Koch [20] to obtain the *N^4^*-methyl- and *N^4^,N^4^*-dimethylcytidine-modified oligonucleotides **44a**–**b** (Scheme 10B). Using the “in solid-phase” approach, the fully protected CPG-linked precursor oligomer **43** (prepared in trityl-on mode) was treated with gaseous methylamine or aq. 40% dimethylamine at 40–65 °C for approximately 2 h. The displacement conditions enabled the effective substitution of the triazolyl group by alkyl and dialkyl amines, and the simultaneous removal of base-labile protecting groups and support cleavage (optionally, alkaline deprotection was performed with aq. ammonia for 30 min at 40 °C when dimethylamine converting reagent was used [20]). After desilylation, the crude RNAs were purified by RP HPLC and detritylated. The RNAs **44a**–**b** were again purified by RP HPLC and analyzed by ESI mass spectrometry. It was found that *N^4^*-monomethylated cytidine-containing oligomer **44a** does not affect the hybridization affinity, while the dimethylated cytidine analog **44b** significantly reduces the binding affinity to target RNAs and can be applied in negative miRNA control [78]. In addition, the *N^4^,N^4^*-dimethylcytidine-modified RNA was used to identify critical 23S rRNA interactions in drug-dependent ribosome stalling [20].

### 3.2. Carbon–Carbon Bond-Forming Reaction via Sonogashira and Stille Couplings

Palladium-catalyzed Sonogashira and Stille cross-coupling reactions between iodonucleobase-modified oligomers and terminal alkynes or unsaturated stannanes, respectively, were employed for the post-synthetic functionalization of RNA by the formation of the new C-C bond. Contrary to the standard protocol of the post-synthetic approach, which involves the full-length oligonucleotide as substrate, the Sonogashira Pd-catalyzed cross-coupling reactions are performed on a column [79], after the incorporation of iodonucleoside phophoramidite. It means that the synthesis is interrupted after iodonucleoside insertion (without deprotection of the 5′-hydroxyl group), and the column is removed from the synthesizer. Then, the reagent mixture for cross-coupling (ethynyl reagent, CuI and Pd catalyst) is loaded to the column, and after 2–3 h of coupling, the column is washed and reinstalled on the synthesizer. The solid-phase synthesis is continued to obtain the desired full-length oligonucleotide, which is followed by its deprotection and release from resin. This on-column method has been found to present several advantages over solution derivatization such as a low consumption of ethynyl reagent, efficient Pd cross-coupling reaction, and a simple washing step to separate reagents from the support-linked oligomer.

The post-synthetic Sonogashira coupling reactions were extensively exploited in the Engels group to construct the RNA oligomers with nitroxide spin labels [65,80,81,82] and pyrene fluorophores [83,84]. The authors described the attachment of spin label 2,2,5,5-tetramethylpyrrolin-1-yloxyl-3-acetylene (TPA) to 5-iodouridine or 2-iodoadenosine (both introduced as a 5′-*O*-DMTr-2′-*O*-TBDMS-protected phosphoramidites) by on-column derivatization using deoxygenated solutions of Pd(0)(Ph_3_P)_4_, CuI in DMF/TEA [80], or Pd(II)(PPh_3_)_2_Cl_2_, CuI in DCM/TEA [65,81]. However, the conditions typical for 5′-*O*-DMTr-3′-*O*-TBDMS phosphoramidite chemistry turned out to be harmful for TPA modification, particularly iodine in pyridine/water (used in oxidation step) and acidic conditions of detritylation. To eliminate this problem, the 5′-BzH-2′-*O*-ACE-protected monomeric units were used (Scheme 11). The combination of the mild *tert*-butylhydroperoxide oxidant with the neutral fluoride for 5′-*O*-BzH removal significantly improved the stability of the nitroxide spin label and increased the efficiency from 4–10% for TBDMS to 35–50% for ACE chemistry [65,82]. The ACE-protected phosphoramidites of 5-iodouridine, 5-iodocytidine, and 2-iodoadenosine were incorporated into 12-, 15-, and 16-nt RNA oligomers, on a 0.2 µmol scale synthesis. The key reaction of Sonogashira cross-coupling involved iodo-RNAs **45**–**47** (Scheme 11), TPA **48**, and a solution of Pd(II)(PPh_3_)_2_Cl_2_ and copper (I) iodide in DCM/TEA (3.5:1.5 *v*/*v*) (the cross-coupling was performed twice to obtain quantitative yields). After 2.5 h, the column was reinstalled on the synthesizer to accomplish the synthesis of the RNA oligomer. Fully assembled RNA was subjected to deprotection and purified by anion-exchange HPLC. 

The combination of ACE phosphoramidite chemistry and Sonogashira coupling was applied for the preparation of pyrene-modified RNA oligomers **53a**–**b** and **54a** (Scheme 12) [83,84]. Two fluorophores, 1-ethynylpyrene **52a** and 1-(p-ethynyl-phenylethynyl)-pyrene **52b**, were attached by reaction with 5-iodocytidine- or 2-iodoadenosine-containing oligomers **46**, **47** (prepared as 10- or 12-nt oligomers on a 0.2 µmol scale) and the solution of Pd(0)(PPh_3_)_4_, CuI in DCM/TEA by on-column synthesis. The coupling was performed twice for 2.5 h each to ensure quantitative reaction. Full-length oligomers were deprotected using a standard order of reactions. The crude oligomers were purified by IE HPLC, affording pure products **53a**–**b** and **54a**.

The on-column concept was applied by the group of Engels to the post-synthetic C–C bond-forming reaction via palladium-mediated Stille cross-couplings (Scheme 13) [85]. The feasibility of Stille coupling was tested for 5-iodouridine- and 2-iodoadenosine-RNA precursors prepared as 5- or 12-mers in three strategies: 2′-*O*-ACE (**46**, **47**), 2′-*O*-TBDMS (**55**, **57**), and 2′-*O*-TC (**56**) on 0.2 or 1 µmol scale. Iodo-oligomers were coupled with several tributylstannanes (RSnBu_3_, **58a**–**d**) such as vinyl-, furyl-, thienyl-, and benzothienyl(tributyl)-stannane. In contrast to Sonogashira post-synthetic cross-coupling, the Stille reaction was performed on column after the completion of automated RNA synthesis. The synthesizer column with fully assembled RNA was connected with the column-syringe system that loaded the solution of tris(dibenzylideneacetone)dipalladium(0) (Pd_2_(dba)_3_, 15 equiv) and tri-2-furylphosphine (P(furyl)_3_, 30 equiv) in dry DMF and then a solution of alkylstannane (60 equiv) in DMF. The column with reaction solution was placed in the oven at 60 °C for 2–18 h. Then, the support-linked modified RNA (still closed in the synthesizer column) was washed, dried, and subjected to deprotection. The ACE-protected, polystyrene-bound oligomers were deprotected using the standard protocol. The TBDMS-protected, CPG-linked oligomers were treated with conc NH_3_/EtOH (3:1 *v*/*v*, 35 °C, 18–24 h) to cleave the base-labile protecting groups and resin, and then, they were desilylated. The TC chemistry was tested exclusively for 5-iodouridine-RNA. The CPG-linked/TC-protected oligomer enclosed in the column was treated with 20% Et_2_NH/MeCN (3 min, rt) to deprotect the phosphate groups. The residual blockage and CPG resin were cleaved with 50% solution of ethylenediamine in dry toluene (2 h, rt), and finally, fully deprotected RNA oligomer was eluted from the column with 0.1 M TEAA buffer. All obtained oligomers **59a**–**d** and **60a**–**d** were purified by IE HPLC. In a few cases, the desired products of Stille cross-coupling reactions were accompanied by reduced deiodinated RNAs or methylated side products.

### 3.3. Cycloaddition Reactions

Two types of post-synthetic cycloadditions were employed in the synthesis of nucleobase-modified oligoribonucleotides, Cu(I)-catalyzed cycloaddition of azides and terminal alkynes (CuAAC strategy) and inverse electron demand Diels-Alder cycloaddition (iEDDA).

The CuAAC reaction (pioneered by Rolf Huisgen [86] and optimized by Sharpless [87] and Meldal [88]) has been exploited for the post-synthetic incorporation of 1,4-disubstituted 1,2,3-triazoles to RNA oligomers by the reaction of ethynyl-modified oligoribonucleotide with an azido-containing reagent. The reverse substrate system is not preferred because azido-modified RNA oligomers interfere with the chemical conditions during automated RNA synthesis, particularly with P(III) form. The representative sites for direct attachment of the ethylene group are positions C7 in the 7-deazapurine and C5 in the pyrimidine precursor nucleosides [50,89,90]. In some cases, the terminal alkyne group was attached to the nucleobases via a linker attached to the exoamino group of guanosine or N3 nitrogen of uridine [35,91,92].

In 2010, the Beal’s group applied a post-synthetic CuAAC strategy to introduce *N*^2^-modifed 2-aminopurine ribonucleosides into an RNA chain to design RNA constructs for siRNA technology (Scheme 14) [35,91]. A 2-propargylaminopurine ribonucleoside was converted to 5′-*O*-DMTr,2′-*O*-TBDMS *β*-cyanoethyl phosphoramidite and incorporated into 12- and 21-nt RNA oligomers by solid-phase synthesis. After deprotection, alkynyl oligomer **61** was conjugated with 2-azido-2-deoxy-D-mannose (**62a**), *N*-azidoacetyl-D-mannosamine (**62b**), 3′-azidothymidine (AZT, **62c**), or azide of *N*-ethylpiperidine (**62d**). Click products **63a**–**d** were generated by incubating an aqueous solution of RNA oligomer **61** with the water-soluble tris-[1-(3-hydroxypropyl)-1*H*-[1,2,3]triazol-4-yl)methyl]amine Cu(I)-stabilizing ligand (THPTA), CuSO_4_, and sodium ascorbate (system for an in situ generation of catalytically active Cu(I) species), and azide **62a**–**d** (5 equiv) for 4 h at ambient temperature. Complete conversions were observed for primary azides **62b,62d**; however, the use of secondary azides **62a**,**62c** reduced conversion yields by 50%. Click products **63a**–**d** were purified by denaturing PAGE and confirmed by ESI-MS. Azide **62b** was also effectively employed for the post-synthetic preparation of the di-click 21-nt RNA oligomer. Thermal denaturation studies indicated that oligomers **63b**–**d** showed similar base pairing stability and specificity to adenosine-containing counterparts. [35,91]. Click products **63b** and **63d** displayed reduced binding affinity to off-target proteins (RNA-dependent protein kinase and adenosine deaminases), while RNAi activity was maintained [35]. 

Further studies of the Beal group focused on the isertion of several 7-substituted 8-aza-7-deazaadenosines into 12-nt siRNA constructs by azide-alkyne cycloaddition (Scheme 15) [89]. 7-Ethynyl-8-aza-7-deazaadenosine-containing oligomer **64** was synthesized using standard phosphoramidite chemistry (1.0 mmol scale) with the coupling time prolonged to 25 min for a 7-ethynyl-modified monomeric unit (prepared as 5′-*O*-DMTr-2′-*O*-TBDMS, NH-dmf amidite with TMS blockage on the ethynyl group to prevent partial hydrolysis during the alkaline deprotection step). After deprotection, precursor oligomer **64** was incubated with aqueous solution of THPTA ligand, CuSO_4_, sodium ascorbate, and azide **65a**–**f** for 6.5 h at rt or 35 °C. Azides were dissolved in aq. Tris-HCl buffer with pH 8.0 (**65a**), water (**65b**–**d**), or DMSO (**65e**, **65f**). Click products **66a**–**f** were purified by denaturating PAGE and identified by MALDI-ToF MS. Study of the modification impact on RNA properties revealed a substantial destabilization of RNA duplexes modified with sterically demanding triazoles **66e** and **66f**. Oligomers with less hindrance (**66a**–**d**) indicated only a slight decrease of duplex stability and a little effect of modification on the base-pairing specificity.

Kellner and co-workers employed click chemistry to label unmodified RNAs with fluorescent dyes (Scheme 16) [92]. For successful labeling, the unmodified RNAs **67** (prepared as synthetic RNA oligomers or yeast tRNA^Phe^ in vitro transcript) were incubated with bifunctional azido-bromo-coumarin **68** in denaturating conditions (70% aq. DMSO, pH 8.5, 37 °C, 3 h) to provide products **69** with chemoselectively N3-alkylated uridine. The azidocoumarin-modified RNA **69** was subjected to click conjugation with fluorescent alkyne-containing AlexaFluor 647 (**70a**) or fluorescein (**70b**) in buffer of pH = 8, in the dark, using Cu^+^ (generated in situ from sodium ascorbate and Cu^2+^) with a THPTA ligand (rt, 2 h). Click products **71a**–**b** were analyzed by PAGE electrophoresis.

The post-synthetic CuAAC reaction was employed by Kerzhner and co-workers to introduce the nitroxide-spin labels into RNA oligomers for EPR/PELDOR measurements [50] (Scheme 17). The 5′-*O*-DMTr-5-ethynyl-2′-deoxyuridine phosphoramidite (additionally protected with a triisopropylsilyl group on ethynyl moiety) was effectively introduced to an RNA oligomer by conventional solid-phase synthesis. Using the “in-solution” approach, the fully deprotected oligomer **72** was conjugated with azide-functionalized nitroxide **73** with CuI instead of a Cu^2+^-ascorbate system (due to the reducible nature of nitroxide radicals) with the excess of THPTA in DMSO-H_2_O solution for 30 min at 37 °C. The spin-labeled oligomer was purified by HPLC to afford the product **74** in total 72% yield. The same oligomer spin labeling was demonstrated by Kerzhner and co-workers in solid phase [90]. In this case, the total reaction time was considerably longer (24 h), and the conversion yield dropped to 36%.

Recently, Nakamoto and co-workers have demonstrated the use of CuAAC reaction for the incorporation of biotin to oligoribonucleotide modified with a novel bifunctional nucleoside analog containing aryl trifluoromethyl diazirine and ethynyl moieties (Scheme 18) [93]. Precursor oligomer **75** was prepared using 5′-*O*-DMTr-2′-*O*-TBDMS building blocks and a standard protocol of phosphoramidite solid-phase synthesis and deprotection. The ethylene–RNA probe **75** was treated with biotynyl-TEG-azide **76** (80 equivalent) in the presence of sodium ascorbate, CuSO_4_ and tris[(1-benzyl-1*H*-1,2,3-triazol-4-yl)methyl]amine (TBTA) for 1 h at rt. The RP HPLC purified product **77** was identified by MALDI-ToF MS.

Beyond the CuAAC cycloaddition, the inverse electron demand Diels-Alder reaction (iEDDA) was employed for the post-synthetic labeling of norbornene-containing RNA oligomer **78** with disubstituted 1,2,4,5-tetrazine-fluorophore derivatives **79a**–**d** (Scheme 19) [94]. The fast conjugation between the ring-strained olefin group as a dienophile and highly electron-deficient 1,2,4,5-tetrazine as a diene was discovered by the Fox and Hilderbrand groups in 2008 [95,96]. To demonstrate the use of iEDDA strategy in the post-synthetic labeling of RNAs, the norbornene modification was introduced into three different oligoribonucleotides (12-, 21-, and 24-mers) using 5-norbornene-modified 5′-*O*-DMTr-2′-*O*-TBDMS uridine phosphoramidite and standard conditions of RNA synthesis. Deprotected oligomers **78** were conjugated with a 2- or 3-fold excess of tetrazine-fluorophore derivatives **79a**–**d** containing the fluorophore Oregon Green 488 in aqueous solution (rt, 1 h). Cycloaddition products were identified by PAGE and MS analysis. It was indicated that the overall yields of cycloaddition reactions were comparable regardless of norbornene position (the single- or double-stranded regions of RNAs). However, they were dependent on the steric effects of the disubstituted tetrazine derivatives with the RNA oligomer (reduced yields were observed for cycloadditions with tetrazines **79c**,**d**). A bioorthogonal reaction of iEDDA between tetrazine-fluorophore **79a** and 21-nt norbornene-modified RNA **78** (prepared as a siRNA duplex) was effectively performed in mammalian cells. In contrast to already known techniques, the fluorophore can be attached to RNA after the target binding, which minimizes the distortion of RNA transport and the functions in the cells.

### 3.4. Derivatization of Amino-Modified Oligoribonucleotides via Formation of Amide Linkage

Amino-prefunctionalized oligoribonucleotides provide a very important class of substrates for the post-synthetic modification/labeling of RNA oligomers. Site-specifically introduced amino-modified units are usually conjugated via amide bond formation with active esters derived from the label and *N*-hydroxysuccinimide (NHS), tetrafluorophenol (TFP), or *N*-hydroxybenzotriazole (HOBt). The active esters react selectively with an aliphatic amino function in solution, after the deprotection of RNA oligomers, without any risk of side reactions with 2′–OH and aromatic amino groups of nucleobases. Based on this strategy, efficient methods for the incorporation of fluorescent dyes, flavin mononucleotide derivative, as well as nitroxide and diazirine labels into RNA oligomers have been developed for their application in the structural and functional analysis of RNA molecules [69,97,98]. Hydantoin RNA modification has also been reported as an example of intramolecular amide bond formation produced by cyclization within the *N*^6^-threonylcarbamoyl functional group attached to an adenosine precursor [63,64].

In 2011, the group of Muller employed amino-modified RNA oligomers **81a**–**c** and **82a,c** as substrates for post-synthetic amino groups conjugation with fluorescent dyes (Alexa488, Cy5 and ATTO647N) and flavin mononucleotide (FAM) (Scheme 20) [97]. Primary amino groups were attached to the C5 position of uridine or C8 position of adenosine via a linker to increase the post-synthetic conversion yields between amino nucleobases and bulky size labels. Two alkenyl- and one alkynylamino linkers were utilized that varied in terms of length and flexibility (Scheme 20a–c). 

To prepare the amino-modified oligomers **81a**–**c** and **82a,c** the standard phosphoramidite chemistry was used with 5′-*O*-DMTr-2′-*O*-TBDMS-protected building blocks and CPG support (the aliphatic amino group of the linkers was protected with trifluroacetyl). Oligomers were effectively deprotected (conc. NH_3_/MeNH_2_-EtOH, 1:1 *v*/*v*, 2 h, rt, then TEA⋅3HF/DMF, 55 °C, 1.5 h) and purified by PAGE electrophoresis. To attach the fluorescent dyes, precursor oligomers **81a**–**c** and **82a**,**c** were dissolved in borax buffer at an approximate pH 9; then, freshly prepared solution of active TFP ester **83** (for Alexa488) or NHS ester **84** (for Cy5 and ATTO647N) in DMF was added. The labeling reactions were carried out at rt overnight in the dark. To conjugate the flavinmononucleotide derivative, the 5-aminoallyluridine-RNA **81a** was treated with R_4_-NHS ester **84** generated in situ using TSTU condensing reagent and DIPEA in DMF. Oligomers containing Alexa488, Cy5, and flavin mononucleotide derivative were purified using denaturing PAGE (yields 15–54%), while ATTO647N-labeled oligomers were isolated by RP HPLC (yields 40–50%). The structures of target oligomers were confirmed by MALDI-ToF mass spectrometry.

Recently, post-synthetic transformations of amino-modified oligoribonucleotides to photo-crosslinkable diazirine-derivatives have been employed to investigate the regulation of protein-RNA interactions by 6-methyladenosine [98]. For preparing the diazirine-containing oligomer **87a** (Scheme 20), (E)-5-(3-aminoprop-1-en-1-yl)uridine-RNA **81a** was suspended in siRNA buffer (Dharmacon, Lafayette, Colorado, USA), then Na–HEPES (pH 7.5) and a solution of diazirine NHS ester (>60 equiv) in DMSO were added. The reaction was vortexed and incubated at ambient temperature for 2–4 h in the dark. The mixture was purified by RP HPLC and analyzed by mass spectrometry. The developed transformation worked also for 3′-biotynylated 6-methyladenosine-modified amino RNA.

The post-synthetic reaction of amino oligomers with active esters was utilized for RNA site-specific spin labeling with nitroxide radicals for EPR measurements (Scheme 21) [69]. Amino-modified guanosine or uridine were introduced into the terminal position of 10-nt oligomers via linkers attached to the N7 position of G (**89**) or C5 position of U (**90**). The precursor oligomers were prepared by the complementary addressed modification method (oligomer **89**) [99] or by the post-synthetic strategy (oligomer **90**, see Scheme 3). Both amino-modified RNAs were spin labeled with new 2,5-bis(spirocyclohexane)-substituted nitroxide and conventional tetramethyl-substituted nitroxide **91b**. The solution of NH_2_-bearing oligomer **89,90** in HEPES-KOH (pH 9.0) was mixed with the respective derivative of NHS ester nitroxide **91a**–**b** dissolved in DMSO (rt, 2 h). The final products **92a**–**b** and **93a**–**b** were separated by HPLC in approximately 70% of conversion yields.

Two cyclic *N^6^*-threonylcarbamoyladenosines containing a hydantoin ring (ct^6^A and 2-methylthio-analog ms^2^ct^6^A) have been recently discovered at position 37 of anticodon loops in several prokaryotic and eukaryotic tRNA sequences [6,7]. To determine the effect of hydantoin structure on the biological activity of tRNAs, model 17-nt RNA oligomers containing ct^6^A and ms^2^ct^6^A were obtained by the post-synthetic methodology based on the cyclization of linear t^6^A/ms^2^t^6^A precursor nucleosides (Scheme 22) [63,64]. The post-synthetic approach was the method of choice, since the hydantoin ring of ct^6^A/ms^2^ct^6^A turned out to be very susceptible to hydrolysis under conditions of phosphoramidite solid-phase RNA synthesis and deprotection. Oligomers **94a**–**b** were synthesized on CPG support at a 2.5 µmol scale using Ultra Mild 5′-*O*-DMTr-2′-*O*-TBDMS-NH-tac-phosphoramidite building blocks to retain the carbamoyl linkage in t^6^A/ms^2^t^6^A intact. The *L*-threonine residue was protected with TBDMS at the hydroxyl function and 2-(trimethylsilyl)ethyl (TMSE) at the carboxyl moiety in both t^6^A and ms^2^t^6^A phosphoroamidites. For the preparation of **94a**–**b**, both precursor amidites were coupled twice in extended time (15 min for t^6^A and 25 min for ms^2^t^6^A) and anhydrous conditions of alkaline deprotection were applied to prevent the TMSE ester amonolysis (fluorolabile TMSE group was cleaved during desilylation, simultaneously with TBDMS blockage). Fully deprotected t^6^A/ms^2^t^6^A-RNAs **94a**–**b** were purified by IE HPLC and subjected to post-synthetic cyclization with aq. DMF solution of 1-ethyl-3-(3-dimethylaminopropyl)-carbodiimide hydrochloride (EDC·HCl) and 1-hydroxy-1*H*-benzotriazole (HOBt) added in the 5- or 10-fold molar excess to **94a** and **94b** precursor oligomers, respectively. After the complete cyclization (1 h, rt) oligomers **95a**–**b** were isolated by centrifugation through Amicon in 80–90% conversion yields. The ct^6^A/ms^2^ct^6^A-oligomers **95a**–**b** were characterized by MALDI-ToF MS and nucleoside composition analysis.

### 3.5. Transformation of Ester Groups

Recently, a new 5-pivaloyloxymethyluridine (Pivom^5^U) convertible nucleoside has been designed for preparing the 5-aminomethyluridine (R^5^U)-modified RNA oligomers **97a**–**e** (Scheme 23) [62]. The studies were focused on the naturally existing 5-aminomethyluridine derivatives such as nm^5^U, mnm^5^U, inm^5^U, cmnm^5^U, and τm^5^U due to their crucial role in tuning the translation process (all modified uridines are positioned at the first anticodon letter of numerous tRNAs) and poor accessibility to oligomers containing these modifications [100,101]. 

It was found that a sterically hindered pivaloyloxyl group (-OPiv) attached to the C-5,1-pseudobenzylic carbon of uridine exclusively promotes the nucleophilic substitution at this position under treatment with ammonia, primary alkylamines, or amino acids at elevated temperature, leading to the quantitative transformation of Pivom^5^U to 5-aminomethyluridines [62,102].

The 5- and 17-nt RNA precursor oligomers **96** were synthesized by the standard protocol of phosphoramidite chemistry on CPG support using a 5’-*O*-DMTr-2’-*O*-TBDMS system of protection (Pivom^5^U–phosphoramidite were coupled twice). The post-synthetic transformation of Pivom^5^U-modified oligomers **96** was conducted “in solid-phase” after the removal of *β*-cyanoethyl groups (TEA/MeCN, 1:1 *v*/*v*, 20 min, rt), simultaneously with the alkaline deprotection of nucleobases and resin cleavage. For this purpose, the CPG-linked Pivom^5^U-RNA **96** was incubated with nitrogen nucleophiles such as ammonia, primary amines (methylamine or isopentenyloamine), or tetrabutylammonium salts of amino acids in ethanolic or aqueous solutions at 60 °C. To get oligomers modified with 5-aminomethyluridine **97a** and 5-methylaminomethyluridine **97b**, Pivom^5^U-RNA **96** was incubated with aq. 30% ammonia or 8 M ethanolic solution of MeNH_2_, respectively for 1 h at 60 °C. To obtain 5-isopentenylaminomethyluridine-RNA (**97c**), precursor oligomer **96** was treated with 0.4 M solution of isopentenylammonium tosylate (600 equiv) and TEA (6000 equiv) in an EtOH–H_2_O mixture for 20 h at 60 °C. The preparation of cmnm^5^U- and τm^5^U-RNAs required the use of glycine and taurine as alkylamine conversion reagents. To increase their nucleophilic character as well as the solubility in organic solvents, both amino acids were transformed to the tetrabutylammonium salt forms. Then, Pivom^5^U-RNAs **96** were treated with 1.5 M ethanolic solution of amino acid salts (3000 equiv) for 20 h at 60 °C. After displacement reactions, the excess of salts was removed from oligomers **97c**–**e** on C18 cartridge. Finally, all the oligomers were desilylated and isolated by IE HPLC in 60–85% yields. The developed strategy was also applied for the preparation of 5-cyanomethyluridine-modified RNA (cnm^5^U-RNA) with potassium cyanide as a nucleophile; however, the conditions used for the complete conversion of Pivom^5^U (20 h, 60 °C) caused the partial degradation of RNA.

### 3.6. Post-Synthetic Conversions of Sulfur-Containing RNA Oligomers 

#### 3.6.1. Post-Synthetic Conversion of 2-Thiouridine-Modified RNA Oligomers

A thiocarbonyl group in 2-thiouridines (s^2^Us) was found to be a reactive site for two post-synthetic RNA conversions. The first strategy was based on the oxidative desulfuration of s^2^U-containing oligomers leading to the formation of 4-pyrimidinone nucleosides (H^2^U) [61]. The second approach involved the reaction of a nucleophilic substitution between alkyl halide and an s^2^U-oligomer where the sulfur atom acts as a nucleophile agent, providing *S*-alkylated analogs of 2-thiouridine [12].

The strong tendency of 2-thiouridine (s^2^U) to desulfuration under oxidative conditions was reported several times at the nucleoside and oligonucleotide levels [103,104,105]. The loss of a sulfur atom led to the formation of uridine and/or 4-pyrimidinone nucleoside (H^2^U) in a ratio that was dependent on the oxidizer (type, concentration), solvent, pH, and type of 5-substituent attached to the nucleobase. The observation that s^2^U-oligomers were desulfured under conditions mimicking the oxidative stress in a cell [104,105] enhanced the need for a reliable access to H^2^Us-modifed oligomers. The attempt to incorporate an H^2^U phosphoramidite into the RNA chain turned out to be not straightforward by the classical approach because of H^2^U-RNA instability under alkaline conditions [104]. In 2015, Chwialkowska and co-workers published a selective post-synthetic transformation of oligomers modified with 2-thiouridine and 5-substituted 2-thiouridines (R^5^s^2^U-RNA, **98a**–**f**) exclusively to the 4-pyrimidinone-containing derivatives (R^5^H^2^U-RNA, **99a**–**f**, Scheme 24) [61]. Six types of R^5^s^2^U modifications were selected to show the generality of the method: 2-thiouridine (s^2^U, R_2_ = H), 2′-*O*-methyl-2-thiouridine (s^2^Um, R_1_ = OMe, R_2_ = H), 5-methyl-2-thiouridine (m^5^s^2^U, R_2_ = -CH_3_), 5-metoxycarbonylmethyl-2-thiouridine, mcm^5^s^2^U, R_2_ = CH_2_COOMe), 5-methylaminomethyl-2-thiouridine (mnm^5^s^2^U, R_2_ = CH_2_NHCH_3_), and 5-taurinomethyl-2-thiouridine (τm^5^s^2^U, R_2_ = CH_2_NHCH_2_CH_2_SO_3_H). Notably, all selected 2-thiouridines are naturally existing ribonucleosides present at the first (“wobble”) position of numerous tRNA biomolecules.

Precursor oligomers **98a**–**f** were synthesized by phosphoramidite chemistry using a 5′-*O*-DMTr,2′-*O*-TBDMS protection system for the preparation of monomeric units and a special protocol of P(III)→P(V) oxidation and work-up to retain the s^2^-thiocarbonyl function intact [100,101,106,107]. Synthetic R^5^s^2^U-modified precursor oligomers were of varied length (from 11 to 23 nts) with 2-thiouridines located both in single and double-stranded regions. Fully deprotected and purified oligomers **98a**–**f** were subjected to oxidative desulfuration by incubation in 2 mM aqueous solution of KHSO_4_ (Oxone^®^, 10-fold molar excess) at 25 °C for 5–30 min. The reactions were terminated by loading on a C18 Sep-Pack cartridge and a standard desalting process. It was found that more than 30 min exposition of the R^5^H^2^U–RNA products to the oxidizing agent leads to further unidentified oxidative lesions. The products **99a**–**f** (obtained in conversion yields 87–95%) were analyzed by MALDI-ToF mass spectroscopy, indicating the selectivity of the reaction toward the 2-thiocarbonyl function. 

In 2012, highly lipophilic *S*-geranylated derivatives of 5-methylaminomethyl-2-thiouridine (mnm^5^ges^2^U) and 5-carboxymethylaminomethyl-2-thiouridine (cmnm^5^ges^2^U) were identified at the first anticodon position in bacterial tRNAs bearing Lys, Glu, and Gln [9]. To understand the biological role of a unique *S*-geranyl modification, the model 2-geranylthiouridine-RNA oligomer **103** (ges^2^U-RNA, homologue of anticodon arm domain of *E. coli* tRNA^Lys^) was synthesized and characterized (Scheme 25) [12]. Three chemical routes were elaborated for preparing the ges^2^U-RNA oligomer, including two post-synthetic strategies based on the selective geranylation of CPG-linked (Scheme 25A) or released/deprotected 2-thiouridine-modified precursor oligomers (Scheme 25B). The synthesis of the s^2^U-RNA oligomer **100** was performed routinely, using a 5′-*O*-DMTr-2′-*O*-TBDMS protection system and *tert*-butylhydroperoxide as an oxidizing reagent. In the first “in solid-phase” method (Scheme 25A), detritylated, support-linked s^2^U-RNA **100** was deprived of *β*-cyanoethyl groups from phosphate residues (TEA/MeCN, rt, 30 min) and then alkylated with > 200-fold molar excess of geranyl bromide and an equimolar amount of triethylamine in ethanol (3 h, rt). The converted product **101** was subjected to alkaline deprotection and support cleavage under mild alkaline conditions (8 M NH_3_/EtOH, rt, 8 h) to avoid substitution of the thioalkyl group with a nitrogen nucleophile, leading to the formation of isocytidine derivatives [108]. After desilylation, ges^2^U-oligomer **103** was purified by anion-exchange HPLC, affording a geranylated product in a total of 40% yield. 

In the second approach, post-synthetic s^2^→ges^2^ transformation was performed in solution (Scheme 25B) by the treatment of fully deprotected/released s^2^U-RNA oligomer **102** with EtOH-water solution of geBr and TEA. The mixture was shaken vigorously for 3 h at rt. Then, the oligomer was filtered through a NAP-G25 column and purified by IE HPLC to furnish pure ges^2^U-RNA **103** at a total of 67% yield. The structure of ges^2^U-RNA product **103** was confirmed by MALDI-ToF mass spectrometry and nucleoside composition analysis.

#### 3.6.2. Post-Synthetic Conversion of 4-Thiouridine-Prefunctionalized RNA Oligomers

Due to the more nucleophilic character of the sulfur atom in s^4^U than other functional groups in nucleic acids, the selective *S*-labeling was reported for both the entire s^4^U-tRNA molecules [109,110] and s^4^U/s^4^dU-modified oligonucleotides [97,111,112,113]. The post-synthetic functionalization of s^4^U was applied predominantly for the site-directed spin labeling of oligomers to extract the long-range distances in RNA or protein–RNA complexes by EPR and NMR techniques [97,112,113,114]. Two general methods of s^4^U-RNA functionalization with thiol-specific reagents were reported, namely by *S*-alkylation (Scheme 26A) and mixed disulfide formation (Scheme 26B).

s^4^U-precursor oligomers **104** were prepared by the standard phosphoramidite chemistry using 5’-*O*-DMTr-2’-*O*-TBDMS-protected monomeric units and *β*-cyanoethyl protection for s^4^-thiocarbonyl group to retain the thio-function intact during the P(III) → P(V) oxidation step. 

In the studies of post-synthetic appending of nitroxide labels to the thiocarbonyl group of s^4^U residue, the iodo-containing proxyl spin label was employed (Scheme 26A) [112]. Fully deprotected s^4^U-RNAs **104** (30- and 31-nt) were dissolved in 100 mM phosphate buffer (pH 8.0), treated with solution of 3-(2-iodoacetamido)proxyl **105** in H_2_O/EtOH (7:3 *v*/*v*), and incubated overnight at rt in the dark. *S*-Alkylated oligomers **106** were isolated (the NAP-10 column and ethanol precipitation) in 80% yields on average. Similar procedures of s^4^U-labeling were employed by others [49,97,114] to gain models for the structural and functional analysis of RNA.

The fast attachment of nitroxide radical to the s^4^-thiocarbonyl function of s^4^U-RNA oligomers was achieved by applying the methanethiosulfonate spin label reagents **107a**–**c** (Scheme 26B) [97,113,115,116]. “Free” s^4^U-precursor RNAs **104** (23- and 57-nt) were incubated with dithiothreitol (DTT) reducing agent in 100 mM phosphate buffer (pH 8.0) for 30 min-2 h at rt. After the removal of DTT excess, the oligomers were treated with appropriate 1-oxyl-2,2,5,5-tetramethylpyrroline- methanethiosulfonates **107a**–**c** in phosphate buffer overnight at rt to afford products **108a**–**c**. Reagents **107a**–**c** were found to be effective for the thioalkyl transfer to the sulfur of 4-thiouridine with formation of the disulfide tethered group. However, the labile nature of the disulfide bond requires the storage of labeled RNAs at −20 °C to minimize the nitroxide radical detachment. 

#### 3.6.3. Post-Synthetic Formation of Disulfide Crosslinks

The site-specific disulfide crosslinking of RNA represents a powerful tool for studying the structure, folding, and function of RNA by locking RNA into a single folded and homogeneous structure. First, disulfide crosslinks were found in cellular tRNAs between neighboring 4-thiouridines [117]. A synthetic procedure for the site-specific insertion of the intrahelical disulfide crosslink into RNA oligomers was based on the oxidation of thioalkyl-modified nucleobases positioned at the 3′- and 5′-ends of oligomers, which after forming the duplex, were able to crosslink with the opposite strand [66,118,119]. The ethane-thiol tethers were attached to the exocyclic amino group of cytosines (Scheme 27A) or to the N3 atom of uracils (Scheme 27B), which were both introduced to oligomers as *S*-protected mixed disulfides **109** and **112**, respectively. The synthesis of *S*-protected *N^4^*-(2-thioethyl)cytidines-modified oligomers (8-nt model of hairpin) utilized the Verdine convertible nucleoside 1 (see Figure 4) and the displacement reaction with cystamine reagent (details are given in Scheme 3) [66], whereas *S*-*tert*-butyl *N^3^*-(2-thioethyl)uridines were transformed to 5′-*O*-DMTr-2′-*O*-TBDMS-protected phosphoramidite as well as nucleoside-modified CPG resin, after which they were introduced to RNA (12- or 72-nts models of hairpin or yeast tRNA^Phe^, respectively) by standard phosphoramidite chemistry [118,119]. The mixed disulfides **109**, **112** were reduced with DTT under aqueous conditions (2–12 h, rt-55 °C) to obtain the free thiols **110**, **113**. After the removal of DTT excess (precipitation, Qiagen cartridge, or dialysis), dithiol-containing oligomers **110**, **113** were redissolved in 25 mM ammonium acetate [66] or 100 mM phosphate buffer (pH 8.0) [118,119] and exposed to air for 5–20 h at rt to form the disulfide bonds. Denaturing PAGE isolation of the target RNA products **111**, **114** revealed nearly quantitative formation of disulfide crosslinks. Since the formation of disulfide crosslinks is mild, highly selective, and does not alter the RNA structure or folding, it was found to be a valuable method for the structural and biochemical evaluation of RNA molecules. 

The thiol-adenine-containing oligomer homologous with the branched region of pre-mRNA was post-synthetically reacted with benzophenone reagent to indicate photo-crosslinks with proteins under splicing conditions [120]. A two-step strategy was chosen to introduce the benzophenone reagent into pre-mRNA substrates (Scheme 28). Firstly, a reactive thiol functionality was introduced into synthetic 10-nt RNA oligomer **115** using Verdine convertible inosine nucleoside **6** (see Figure 4), which permits the installation of cystein disulfide tether at the N6 position of adenosine (see Scheme 3). After the incorporation of synthetic **115** oligomer into full-length pre-mRNA molecules via the ligation procedure, the disulfide was reduced to thiol-RNA **117** using DTT. The RNA construct **117** was subjected to derivatization with highly effective thiol-specific benzophenone maleimide photoreagent in 50% aq. DMF for 1 h at rt.

The obtained benzophenone-pre-mRNA **118** was incubated in HeLa nuclear extract under standard splicing conditions and crosslinked with proteins by irradiation with a 302 nm lamp.

## 4. Conclusions

Every year, new strategies are discovered to synthesize modified oligonucleotides. Nevertheless, there is still a growing demand for the development of new approaches, particularly toward RNA oligomers, the preparation of which is more challenging than their DNA counterparts. Unlike enzymatic procedures, the chemical synthesis of RNA oligomers offers the possibility of site-specific incorporation of modified nucleosides, making RNA synthesis an important tool for many interdisciplinary studies. The hereby presented concept of post-synthetic oligonucleotide modification represents an important alternative for standard phosphoramidite chemistry, particularly when target modifying groups/labels are unstable or reactive under conditions of solid-phase synthesis. The great advantage of post-synthetic reactions is that one building block with a reactive group can react with a wide variety of reagents, providing several variously modified oligomers of homologous sequences. In addition, reported precursor oligomers can be easily adapted to incorporate new modifications using already well-optimized post-synthetic reactions. Some of the post-synthetic transformations are bioorthogonal and have been successfully employed to study and monitor RNA nucleic acids in their native environment [121,122,123].

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
