# Peer review of "Synthesis of Nucleobase-Modified RNA Oligonucleotides by Post-Synthetic Approach"

_molecules, 2020, doi:10.3390/molecules25153344_

Round 1

Reviewer 1 Report

Comment:  

The authors have written a very professional review, which is focused on postsynthetic modifications of RNA oligonucleotides, the importance of which for a deeper understanding of biochemical and biological processes at the molecular level is constantly growing. The individual types of modifications are described in a clear and readable way. Schemes and Figures are always placed close to the text, so the reader does not have to flip through and laboriously search for the desired data. References are adequate to the topic.
Nevertheless, I have three major comments.

  1. Figure 2 shows commonly used 5’-O-DMTr-ribo-3’-phosphoramidite monomers. I am missing reverse riboamidites (ChemGenes), the 3’-O-DMTr-ribo-5’-phosphoramidite monomers which allow the solid phase synthesis in reverse order, e., from the 5’ to the 3’end. This synthesis in reverse order provided very pure oligoribonucleotides. A short paragraph about those compounds should be implemented into the text.

  1. Scheme 3 has little informative value. It is too general and takes a lot of attention to understand it. I propose to delete Scheme 3. On the other hand, quite perfect are all individual Figures and Schemes with concreete reagents and reactions.

  1. Alphabetic list of abbrewiations should be included. I am convinced that it could help to a diversely oriented readers.

I also found some minor inaccuracies in the text that should be corrected.

Figure 2 e)

Green R in formula should be in black for better orientation

Line 128

replace 4-(tert-butylphenoxy)acetyl with 4-tert-butylphenoxyacetyl

Line 508

replace ..... pirens with .... pyrenes

Line 556

replace N2-Propargylpurine … with  2-Propargylaminopurine …

Line 569 (legend in Scheme 15)

replace  ... N2 -propargyl-2-aminopurine-RNA …  with … 2-propargylaminopurine-RNA …   

Line 622-623

replace   ... tris[(1-benzy-1H-1,2,3-triazol-4-yl)methyl]amine … with … tris[(1-benzyl-1H-1,2,3-triazol-4-yl)methyl]amine …

Line 632

replace  ... Hirderbrand … with  ... Hilderbrand …

Line 690

replace … interactome … with … interaction …

Line 691

replace  … 5-aminoallyluridine-RNA with … (E)-5-(3-aminoprop-1-en-1-yl)uridine-RNA …

Line 846 (legend in Scheme 26)

replace  … 24 godz  with  … 24 h

Conclusion: After minor revision, I recommend this review-manuscript for publication in Molecules.

Author Response

Response to Reviewer 1 Comments

Point 1: Figure 2 shows commonly used 5’-O-DMTr-ribo-3’-phosphoramidite monomers. I am missing reverse riboamidites (ChemGenes), the 3’-O-DMTr-ribo-5’-phosphoramidite monomers which allow the solid phase synthesis in reverse order, e., from the 5’ to the 3’end. This synthesis in reverse order provided very pure oligoribonucleotides. A short paragraph about those compounds should be implemented into the text.

Response 1: In regard to reviewer comment we added a short description of reverse RNA synthesis and attached the structure of proper monomeric unit (see Figure 2). We also added proper literature (ref. 47 and 48). The fragment implemented to the text is indicated in yellow.

Point 2: Scheme 3 has little informative value. It is too general and takes a lot of attention to understand it. I propose to delete Scheme 3. On the other hand, quite perfect are all individual Figures and Schemes with concreete reagents and reactions.

Response 2: Scheme 3 was removed from the paper.

Point 3: Alphabetic list of abbrewiations should be included. I am convinced that it could help to a diversely oriented readers.

Response 3: The list of abbreviation was added (starts from line 993).

Point4: I also found some minor inaccuracies in the text that should be corrected.

Response 4: All indicated inaccuracies were corrected.

Reviewer 2 Report

This manuscript is an excellent review of the development of chemical modifications to enhance RNA.

Page 8- Please add the word oligonucleotides to clarify the numbers 14-16 on page 8, line 258.

Please consider including why modifications were examined. How were they hypothesised to enhance RNA e.g.  yield, stability, enable visualisation, crosslinking to nanocarriers, etc? For example on page 9, line 277-279, “by the Hobartner group was to construct spin-labeled RNA oligomers with nitroxide radicals…” 

An example of when this has been provided for the modification is on page 27, line 886-888, for disulphide cross-links.  This greatly aids the reader in understanding why the modification is beneficial.

This is a point throughout the review, which is rich in technical detail of the modifications, but lacking in the reasoning as to why each modification was developed.

Additionally, it would be beneficial to list the two current RNAi therapeutics approved for use and explain what (if any) modifications are used or could be used to enhance stability etc.

Further a brief mention of how particular modifications could be translated to scaled up current Good Manufacturing Practice to enable clinical trials of RNAi therapeutics would strengthen the review.

Author Response

Response to Reviewer 2 Comment

Point 1: Page 8- Please add the word oligonucleotides to clarify the numbers 14-16 on page 8, line 258.

Response 1: The word oligonucleotides was added

Point 2: Please consider including why modifications were examined. How were they hypothesised to enhance RNA e.g.  yield, stability, enable visualisation, crosslinking to nanocarriers, etc? For example on page 9, line 277-279, “by the Hobartner group was to construct spin-labeled RNA oligomers with nitroxide radicals…” An example of when this has been provided for the modification is on page 27, line 886-888, for disulphide cross-links.  This greatly aids the reader in understanding why the modification is beneficial. This is a point throughout the review, which is rich in technical detail of the modifications, but lacking in the reasoning as to why each modification was developed.

Response 2: In regard to reviewer comment we have slightly expanded the description of the role/properties of modified units incorporated to RNA via post-synthetic strategy and explained why this approach is beneficial compared to classical method. The fragments implemented to the text are indicated in green.

Point 3: Additionally, it would be beneficial to list the two current RNAi therapeutics approved for use and explain what (if any) modifications are used or could be used to enhance stability etc.

Response 3: This information was included in the introduction. The fragment implemented to the text is indicated in green (lines from 55).

Point 4: Further a brief mention of how particular modifications could be translated to scaled up current Good Manufacturing Practice to enable clinical trials of RNAi therapeutics would strengthen the review.

Response 4: In regard to the reviewer suggestion we added a short information about the large-scale synthesis of therapeutic oligonucleotide and proper literature (ref 53). The fragment implemented to the text is indicated in green (lines from 154)

Reviewer 3 Report

The article, proposed by K. Bartosik and al., is an interesting and useful review in the field of modified nucleic acids (RNA).
It reports the main strategies for obtaining base-modified RNA oligonucleotides through reactions post solid-phase ON synthesis.

The review, which covers a good part of the pertinent literature, is well organized, and the Schemes and Tables are clear in describing the discussed reactions.

Furthermore, for each reaction, the authors indicate, albeit minimally, the use that the researcher makes of the modified RNA. This is undoubtedly useful for the reader who works in the field of nucleic acids.
In summary, I believe that this review deserves the publication
on the Journal Molecules.

Author Response

Response to Reviewer 3 Comment

Point 1: Furthermore, for each reaction, the authors indicate, albeit minimally, the use that the researcher makes of the modified RNA

Response 1: In regard to the reviewer comment we have slightly expanded the description of the role/properties of modified units incorporated to RNA via post-synthetic strategy.